# Corporate Environmental Protection Behavior and Sustainable Development: The Moderating Role of Green Investors and Green Executive Cognition

**DOI:** 10.3390/ijerph20054179

**Published:** 2023-02-26

**Authors:** Jie Zhou, Shanyue Jin

**Affiliations:** College of Business, Gachon University, Seongnam 13120, Republic of Korea

**Keywords:** corporate environmental responsibility, environmental protection investment, sustainable development, green investors, green executive cognition

## Abstract

Faced with serious environmental problems, companies have become important participants in environmental protection efforts. By assuming environmental responsibilities and pursuing environmental protection, enterprises can create a good image, gain public and government support, and expand their influence. Simultaneously, green executive cognition and green investors play important roles in enterprises and the market economy. This study examines whether the environmental protection behavior of enterprises has a positive impact on their sustainable development, and how green investors and green executive cognition affect the relationship between environmental protection and sustainable development. This study adopts a fixed effects regression method to research Chinese A-share listed companies in 2011–2020. The results show that enterprises’ performance regarding environmental responsibilities or investment promotes sustainable development. The higher the participation of green investors or the higher the awareness of green executives, the more the environmental responsibility performance and environmental investment promote enterprises’ sustainable development. This study enriches the literature on the environmental protection behavior of enterprises and the sustainable development of enterprises as well as provides a theoretical foundation for related research. Moreover, the role of green investors and green executive cognition in promoting environmental protection and the sustainable development of enterprises will inspire investors and executives.

## 1. Introduction

China became the world’s second-largest economy in 2010 [1]. However, its rapid economic growth has resulted in serious environmental problems [2]. Because enterprises are the largest participants in the economy and use the most natural resources, many companies have continued to seek greater economic benefits at the expense of long-term environmental damage, making the environmental threats facing China much more serious. Therefore, companies are responsible for environmental issues [3]. To resolve the contradiction between environmental protection and economic development, enterprises must focus on environmental protection. Corporate environmental behavior plays a vital role in achieving the dual goals of improving environmental quality and promoting sustainable economic development.

As environmental sustainability is becoming increasingly important for economic development, many Chinese companies have begun to implement environmental protection measures. Enterprises have begun to enhance their ecological consciousness, carry out environmental protection activities, and increase their investments in environmental protection, while actively assuming environmental responsibility [4]. With the improvement in global environmental protection awareness, managers, companies, governments, and other stakeholders are paying increasing attention to enterprises’ social and environmental responsibilities [5]. The question is, will involvement in environmental protection significantly affect enterprises’ sustainable development?

First, corporate environmental responsibility (CER) has become an international trend. This trend is consistent with the goals of economic development and environmental protection. According to the concept of sustainable development, enterprises’ profits and environmental protection are compatible [6]. A company’s profitability and the implementation of environmental responsibilities are not necessarily mutually exclusive goals. Enterprises can achieve sustainable development through business behavior that shows concern about the environment. Therefore, the profits generated by companies involved in environmental protection should be sustainable. Environmentally responsible acts can make a company sustainable and simultaneously provide companies with additional resources and competitive advantages [7]. In addition, enterprises’ responsibility for the environment will increase their reputation, making them an attractive investment. Such a company can obtain the support of stakeholders, thereby reducing operating risks and achieving long-term stable growth [8]. In other words, enterprises may sacrifice certain economic benefits in the short term but guarantee sustainable development in the long term [9]. Academics and industries have reached a consensus on the significance of CER. It is believed that the environmental responsibility of enterprises is an indispensable link in the process of achieving sustainable development [10].

Second, environmental investment is an investment activity with environmental protection as its main purpose, and it can lead to improved environmental performance. Enterprises can achieve corporate value and sustainable development through environmental protection investment activities [11]. Enterprises have increased their environmental protection investment, earnestly fulfilling their responsibility towards energy conservation, emissions reductions, and environmental protection; they have also invested funds for carrying out environmental pollution control. This can convey positive information to society, help speed up the shaping of the corporate image, and enhance consumers’ recognition of corporate products [12]. With the continuous increase in environmental protection investment, although it will bring economic pressure to enterprises in the short term, the increase in clean innovation and reduction in the environmental cost of enterprises [13] will realize the sustainable utilization of resources, effectively reduce environmental pollution, and contribute to the sustainable development of enterprises.

In addition, green investors, as fund investment entities that take environmental and social responsibilities into account, must comprehensively consider multiple factors, such as economic, social, and environmental performance during the investment process, and play strong supervision and governance roles in the development of enterprises [14]. The economy will benefit from the dual effects of income and environmental protection [15]. When green investors choose investment objects, whether the project meets environmental testing standards, pollution control effects, and ecological protection are important prerequisites, thereby achieving the purpose of advanced governance [16]. Therefore, green investors can promote enterprises’ investment in environmental protection, help improve their environmental quality, and rapidly transition to a sustainable growth state [17].

At the same time, due to the upward trend in environmental awareness, organizers of organizational policies are more inclined to implement green practices to achieve their environmental sustainability goals [18,19,20]. As makers of corporate strategic decisions, executives’ responses to environmental changes are affected by their own awareness of environmental issues [21], and executives integrate green cognition into their daily management activities, which is conducive to improving the company’s environmental implementation power. This guides executives to pay attention to ecological environmental protection issues when making decisions and promotes the sustainable development of enterprises [22].

Currently, green development is the primary task of many countries. As the main force behind economic development, enterprises are not only responsible for market demand but also for the protection of resources and the environment [23]. Therefore, the core issue for the future development of enterprises is to maintain the long-term stable development of enterprises to protect the environment [24]. This study analyzed the relationship between a company’s participation in environmental protection and sustainable development through a combination of theoretical and empirical analyses as well as explored whether the cognition of green investors and executives affects the participation of enterprises in environmental protection and sustainable development. This study aims to analyze the behavior of enterprises participating in environmental protection from two aspects: CER and environmental protection investment. This study explores the role of corporate environmental protection behavior in achieving the dual goals of improving environmental quality and promoting sustainable economic development.

The contributions of this study are as follows. First, this study divides enterprise environmental protection behavior into two parts: environmental investment and environmental responsibility, enriches the literature on the environmental protection behavior of enterprises and the sustainable development of enterprises, and provides a theoretical foundation for research in related fields. Second, this study explores the role of green executives’ cognition in promoting the relationship between environmental protection and corporate sustainable development from within enterprises as well as provides insights into how top managers within enterprises can promote the green behavior of employees and embed green culture in the entire organization. Furthermore, outside of the enterprises themselves, executives’ actions can increase the number of green investors and influence government policies and regulations, which are key to driving corporate green investment. Thus, this study provides new insights into the formulation of environmental policies. Third, the research results of this study provide important practical implications for companies to design sustainable development strategies. In the long run, the environmental protection behavior of enterprises can improve environmental quality, promote economic development, enhance enterprise value, and help enterprises establish long-term competitiveness.

## 2. Theoretical Background and Hypotheses

### 2.1. Enterprise Environmental Protection Behavior and Sustainable Development

Environmental economics provides an indispensable theoretical foundation for studying environmental responsibility. The theory of environmental economics proposes that economic development depends on the development of the ecological environment, and it is necessary to grasp the balance of and coordination between the environment and economy. The theory of environmental economics emphasizes that, while meeting the increasing material needs of people, it is necessary to consider the relationship between economic development and the environment, coordinate the relationship between man and nature, and always take ecological balance as a prerequisite for corporate development [25]. Therefore, enterprises need to fulfill their own environmental responsibility to promote sustainable development. Enterprises bear environmental responsibilities and need to use relevant theories in environmental economics, such as environmental evaluation methods, analyses of environmental cost benefits, and environmental protection economic analyses, to make decisions that are beneficial to enterprises and the environment, realize a win–win situation for enterprises and the environment, and promote the sustainable development of enterprises [26].

According to signal theory, a company can use CER as a signal to convey a positive image to the public. CER can help a company build a good reputation in the eyes of different stakeholders, obtain legitimacy and resources from stakeholders, and obtain the support of the government and the community to enhance its performance [8]. Enterprises are increasingly regarded as basic elements of strategic management and are used as driving forces to improve corporate sustainability [27]. Enterprises’ performance in terms of environmental responsibilities also helps improve corporate competitiveness, break through trade and market barriers, and achieve the sustainable development of enterprises [28]. Related studies have shown that listed companies can not only alleviate financing constraints by fulfilling their corporate environmental responsibilities [29], but also reduce their tax burden [30], further promoting corporate performance. Management determines three elements of sustainable development: innovation, standardization, and rationality. Innovative sustainable development relies on the concept of ecological efficiency. Ecological innovation is implemented to achieve economic advantages, improve the efficiency of resource use (materials and energy), reduce emissions and costs, and promote the sustainable development of enterprises [31]. Studies have shown that a company can increase its CER participation to enhance its competitive advantage and increase its value by strengthening its innovation [6]. A direct embodiment of an enterprise’s positive attitude toward environmental protection is fulfilling its environmental responsibility. Enterprises that are willing to take the initiative to assume environmental responsibility usually have an advantage in achieving sustainable development [32].

In summary, CER can help enterprises pay attention to the environment, enhance the performance of the company, increase its value, and promote sustainable development. Therefore, this study proposed the following hypothesis:

**Hypothesis** **1.***Enterprises’ performance regarding environmental responsibilities has a positive impact on their sustainable development*.

From the perspective of the basic theory of resources [33], environmental protection, as a key factor in performing social responsibilities, can also be regarded as an important resource for enterprises. Enterprises usually take the initiative to invest in environmental protection because environmental investment is conducive to improving the social benefits of the enterprise, increasing the government’s recognition of the enterprise and its legitimacy, improving corporate financing capabilities, and obtaining tax reduction support [34].

Enterprise environmental investment is an important aspect of corporate environmental strategies. From the perspective of specific enterprises’ environmental investment performance, there are many aspects related to environmental protection investment by enterprises, such as greening, pollution fees, environmental protection technology certification fees, and environmental technology development fees [35]. Some scholars believe that enterprises can establish a corporate image with good environmental awareness in the market by conducting environmental protection investments in the market and enhancing corporate performance. Therefore, we need to pay attention to the efficiency of enterprises’ environmental investments in sustainable development [36]. Simultaneously, corporate environmental investment also helps improve corporate environmental performance. This is an effective micro-solution to alleviate the environmental problems generated by enterprises’ excessive resource and energy use [37]. Additionally, investment in society and the environment is becoming a benchmark for the financial market. Research shows that enterprises’ investment in environmental protection can improve their efficiency and enhance their sustainability [24].

In summary, enterprises carrying out environmental protection investments can address corporate environmental problems, effectively carry out environmental protection, and promote the sustainable development of enterprises. Therefore, this study formulated the following hypothesis:

**Hypothesis** **2.***Enterprises’ environmental protection expenditure has a positive impact on their sustainable development*.

### 2.2. The Regulatory Role of Green Investors

From the perspective of institutional investors, because of the influence of social norms and moral constraints, they tend to invest in enterprises with good credit, high social responsibility, and a green bond market [15]. This helps improve corporate environmental governance and social responsibility, which enhances the value of the enterprise itself [38,39]. As institutional investors, green investors pay attention to environmental responsibility goals and achieve higher financial performance [40].

Studies have found that investors with environmental awareness remain consistent with their morality, beliefs, and investment choices through the application of various screening methods. Companies transitioning to green investments will increase their environmental management during daily operations [41]. Although some scholars associate green investment with higher costs and negative impacts on a company’s profits, some studies have shown that green investment can accelerate the growth of corporate profits and cost savings [42]. Green investors can encourage enterprises to implement green behaviors, fulfill corporate responsibilities, increase environmental protection expenditure, and promote their sustainable development. Therefore, this study developed the following hypotheses:

**Hypothesis** **3.***Green investors play a positive adjustment role in the impact of CER on the sustainable development capacity of enterprises*.

**Hypothesis** **4.***Green investors play a positive adjustment role in the impact of corporate environmental protection expenditure on the sustainable development capacity of enterprises*.

### 2.3. Regulatory Role of Green Executive Cognition

Senior theory states that the internal and external environments faced by executives are complicated and often include difficulties in understanding. As decision-makers of corporate actions and strategies, the cognitive foundations and values of executives limit their ability to interpret relevant information. Therefore, the characteristics of the cognitive ability, perception, and values of executives affect the strategic choices and performance of the organization [43,44]. Executives make strategic choices for a company based on their own cognition and values [45]. Therefore, the green cognition of executives is the perception of resource and environmental issues based on their own knowledge structure and values [46].

In the strive for continuously improving the environment in China, when senior management recognizes the income from the implementation of green measures, they will be committed to participating in measures that will eventually improve environmental performance [47]. Studies have found that the highest management level can adopt environmental protection methods at all levels of the organization. Awareness of environmental problems and commitment to monitoring a company’s environmental activities can effectively improve the environmental performance of an enterprise [48]. In addition, if an enterprise hires high-level managers with green cognition, it may enhance its competitive advantage in CER. Therefore, green managers are more responsible for the environment and a company’s sustainable growth [23]. Green executive cognition can encourage enterprises to participate in environmental protection, fulfill their environmental responsibility, increase their environmental protection expenditure, and achieve sustainable development. Therefore, this study proposed the following hypotheses:

**Hypothesis** **5.***The green cognition of executives plays a positive adjustment role in the impact of CER on the sustainable development capacity of enterprises*.

**Hypothesis** **6.***The green cognition of executives plays a positive adjustment role in the impact of corporate environmental protection expenditure on the sustainable development capacity of enterprises*.

Figure 1 is the research model.

## 3. Research Methods

### 3.1. Data and Samples

Considering the influence of the financial crisis in 2008 and the release of the Environmental Responsibility Assessment System by Hexun in 2010, this article first selected Chinese A-share listed companies from 2011 to 2020 for this research paper. In order to enhance the reliability of the paper, samples were removed according to the following conditions: first, listed companies in the financial industry; second, non-regular trading companies, such as special treatment (ST; which means listed companies with negative net profit for two consecutive fiscal years), ST* (which represents a delisting warning due to the loss of listed companies for three consecutive fiscal years), PT (particular transfer, which means listed companies that stopped any transactions, cleared the price, and waited to be delisted), and delisted companies; and third, samples that were seriously lacking in abnormal observation values and data. The study finally obtained 6545 sample observation values. To avoid the resulting extreme values affecting the results, this study performed a 1% level retractable treatment of all continuous variables. The data used in this study were from the Hexun Social Responsibility Assessment System, WIND Database, and China Stock Market and Accounting Research Database (CSMAR). The diversified regression analysis used the STATA 16.0 software.

### 3.2. Definition and Measurement of the Variables

#### 3.2.1. Sustainable Development Capabilities

Sustainable development refers to the sustainability by which enterprises can ensure profitability and competitive advantages in the process of pursuing their business goals. In the current research, there are many ways to measure the sustainable development capabilities of enterprises. This study used James C. Van Horne’s static model to measure an enterprise’s sustainable development capacity from the perspective of corporate profitability and competitive advantage [49]. The formula for the SGR index is as follows:SGR = net sales interest rate × total asset turnover rate × income reservation rate × equity multiplication/(1 − net sales interest rate × total asset turnover rate × income reservation rate × equity multiplication).

#### 3.2.2. Enterprise Environmental Responsibility

This study adopted the environmental responsibility score in the social responsibility report provided by the China Hexun website as a measurement indicator [45,50]. The environmental responsibility score is mainly based on five indicators: environmental awareness, environmental management system certification, environmental protection investment cost, number of pollutant emissions, and types of energy conservation. The system can comprehensively and objectively score the environmental responsibility performance levels of listed companies. The score ranges from 0 to 30 points; the higher the score, the more environmentally friendly the production and operation of the listed company, and the higher its enthusiasm for performing environmental responsibilities [51].

#### 3.2.3. Enterprise Environmental Protection Expenditure

To reduce the impact of subjectivity, we extracted data from the notes of corporate financial statements [52,53]. Through filtering the financial statements of the listed companies, keywords related to environmental protection investment in the construction of the financial statement, other keywords related to environmental protection investment, and relevant environmental investment data were obtained. However, the two ends of the asset-liability statement loan (under construction and management expenses and other payments payable) were directly added to the duplicate calculation. Therefore, only two items of construction and management costs were selected for screening. The keywords for extracting and screening included cleaning, greening, environmental protection, sewage, energy saving, carbon dioxide, and emissions reduction. Because of the independent third-party audit of the financial statements of the listed company, the enterprise’s environmental investment was calculated and environmental protection-related keywords were extracted from the financial statement notes, making the environmental protection investment data more objective and real. To improve the stability of the data, we used the natural number of environmental investments by the company that year [11,54].

#### 3.2.4. Green Investors

Since green investors are institutional investors [55], and this study used the CSMAR. First of all, the “fund main information table” in the fund market series and “stock investment details” were matched so as to obtain a fund details table for investing in listed companies. Second, the “investment goals” and “investment scope” of each fund were manually queried using “environmental protection”, “ecology”, “green”, “new energy development”, “clean energy”, “low-carbon”, “sustainable”, and “energy saving”. If these environment-related words were present, it was assumed that the company had green investors and a value of 1 was assigned, otherwise 0 was assigned.

#### 3.2.5. Green Cognition of Senior Executives

Text analysis has been proven to effectively measure executive cognition and can be used in vertical data research [56]. The data required for measuring executive cognition came from the annual reports of the listed companies. Therefore, this study adopted a text analysis method. Based on green competition advantages, corporate social responsibility cognition, and external environmental pressure, three dimensions were selected for a series of keywords. Using the above words, the annual reports of A-share CSI-listed companies in 2011–2020 were searched. The green cognition of executives was mentioned frequently. To improve the stability of the data, we used the frequency of the annual reports of listed companies.

#### 3.2.6. Control Variables

To prevent other factors from interfering with the research results [57], discovery, enterprise size (size), asset-to-liability ratio (LEV), cash flow ratio (cash flow), company development (GROWTH), total asset turnover (ATO), enterprise age (firm age), and the influence of the year variable (year) and industry variables (industry) were controlled. Table 1 lists the variables and their measurements.

### 3.3. Model Design

To verify the hypotheses and assumptions of this study, we used the regression model of the fixed effects of CER and environmental investment on the sustainability of the enterprise (Model (1)) and (Model (2)).

*β*1 was positive and significant in Model (1), indicating that companies with more environmental responsibilities obtained more sustainable development capabilities. *β*1 was positive and significant in Model (2), indicating that enterprises can gain higher sustainable development capabilities for environmental protection investments.

To verify the regulation of the relationship between environmental protection and the sustainable development capacity of an enterprise with awareness of green executive cognition, we built a model (Equations (3) and (4)).

In Equations (3) and (4), if *β*3 is greater than zero and significant in the model equation, it indicates that green executive cognition has a positive adjustment effect on environmental protection and sustainable development capabilities.

To verify Hypotheses 5 and 6 regarding green investors and the relationship between environmental protection and sustainable development capabilities, we built two models (Equations (5) and (6)).

In Equations (5) and (6), if *β*3 is greater than 0 and significant, it indicates that green investors have a positive adjustment effect on environmental protection and sustainable development capabilities.

Equations (1)–(6) show the models of the study.
(1)GR=β0+β1CER+β2SIZE+β3LEV+β4CASHFLOW+β5GROWTH+β6ATO+β7FIRMAGE+∑YEAR+∑IND+ε  
(2)SGR=β0+β1EPI+β2SIZE+β3LEV+β4CASHFLOW+β5GROWTH+β6ATO+β7FIRMAGE+∑YEAR+∑IND+ε
(3)SGR=β0+β1CER+β2GEC+β3CER∗GEC+β4LEV+β5CASHFLOW+β6GROWTH+β7ATO+β8FIRMAGE+∑YEAR+∑IND+ε
(4)SGR=β0+β1EPI+β2GEC+β3EPI∗GEC+β4LEV+β5CASHFLOW+β6GROWTH+β7ATO+β8FIRMAGE+∑YEAR+∑IND+ε
(5)SGR=β0+β1CER+β2GI+β3CER∗GI+β4LEV+β5CASHFLOW+β6GROWTH+β7ATO+β8FIRMAGE+∑YEAR+∑IND+ε  
(6)SGR=β0+β1EPI+β2GI+β3EPI∗GI+β4LEV+β5CASHFLOW+β6GROWTH+β7ATO+β8FIRMAGE+∑YEAR+∑IND+ε

The *p*-values of the Hausman test results of Models (1)–(6) in this study were all <0.05; therefore, the fixed effect regression model was the most appropriate choice [58].

## 4. Empirical Analysis Results

### 4.1. Descriptive Statistics

The average value of the companies’ sustainable development capacity (SGR) was 0.0614, and the standard difference was 0.0518, which indicated that the level of sustainable development capacity of the sample companies was generally low and there was a gap; the average CER was 1.208, the maximum value was 20, and the minimum value was 0. This indicates that there was a gap between the environmental responsibility scores of different enterprises and the overall environmental responsibility score of the enterprises was not high; the average value of environmental protection input (EPI) was 15.70, the maximum value was 21.65, and the minimum value was 0, which indicates that the environmental investment in the sample enterprises was high. The standard deviation was 3.356, indicating that there was a gap between the environmental investment in the sample enterprises. The maximum value of green investors (GI) was 1, the minimum value was 0, and the average value was 0.193, indicating that there were few green investors. As for green cognition, the maximum value was 6.098, the minimum value was 0.693, and the average value was 3.326, indicating that the cognitive level of green executives in the enterprises was at a medium level. Table 2 outlines the results.

### 4.2. Correlation Analysis

This study used the Pearson correlation coefficient matrix to analyze the correlation between the models’ dependent and independent variables. As Table 3 shows, there was a significant positive correlation between CER and the companies’ sustainable development capacity as well as between environmental protection investment and the companies’ sustainable development capacity. Similarly, green investors and green executive cognition were significantly positively related to the sustainable development capacity of the companies. Furthermore, the variance inflation factor value of each variable was <3, indicating that there was no multicollinearity problem.

### 4.3. Regression Analysis Results

The results of the regression analysis are presented in Table 4. As shown in column (1), the companies’ environmental responsibility was significantly positive at the 1% level, with a coefficient of 0.0020, indicating that environmental responsibility had a significant positive impact on the sustainable development capacity of the enterprises. Therefore, Hypothesis 1 was supported. As shown in column (2), environmental protection investment was significantly positively related to the sustainable development ability of the enterprises at the 1% level, with a coefficient of 0.0026, indicating that environmental protection investment had a significant positive impact on the sustainable development ability of the enterprises. Thus, Hypothesis 2 was supported.

The interactive items of the adjustment variables and independent variables were added based on the regression model to test the adjustment effect. The interaction coefficient of green executive cognitive and CER, shown in column (3), was positive at the 1% level, which supported Hypothesis 3. As shown in column (4), the interaction term coefficient between green executive cognition and environmental protection input was positive and significant at the 5% level, supporting Hypothesis 4. The interaction term coefficient of green investors, shown in column (5), was positive and significant at the 1% level, supporting Hypothesis 5. As shown in column (6), the interaction term coefficient between green investors and environmental protection investment was positive and significant at the 1% level, thus supporting Hypothesis 6.

The empirical results showed that green executive cognition and green investors had a positive effect on enterprises fulfilling their environmental responsibility and enterprises’ high environmental protection investment, which can strengthen their participation in environmental protection and have a positive impact on their sustainable development ability, forming a green cycle.

### 4.4. Robustness Tests

Considering the possible endogeneity problem caused by the omitted variables and bidirectional causality, as well as other factors, and the possible estimation bias brought on by this endogeneity problem, this study referred to the practice of Zhang et al. [58] and selected the one-period lag of the environmental responsibility score as an instrumental variable [59] and used the two-stage least squares (2SLS) method to test the robustness.

The regression results for the 2SLS method are shown in columns 1–4 in Table 5. In the first stage (columns 1 and 3), the regression coefficients of L.CER and CER were significantly positive at the 1% level (0.2182 ***) and L.EPI and EPI were significantly positive at the 1% level (0.2093 ***). In the second stage (columns 2 and 4), the regression coefficients between the CER score after fitting by L.CER in the first stage and the SGR were also significantly positive at the 1% level (0.0042 ***); the regression coefficients between the EPI score after fitting by L.EPI in the first stage and the SGR were also significant at the 1% level (0.0055 ***). These results show that after considering the endogeneity problem, the enterprise environmental responsibility score and environmental protection investment were still significantly and positively correlated with the sustainable development capacity of enterprises, which once again verified Hypotheses 1 and 2.

In addition, with regard to the 2SLS test, the under-identification test (the Kleibergen-Paap rk LM statistic) for Models (1) and (2) was 22.633, and for Models (3) and (4) it was 19.533, with a corresponding *p*-value at 0.0000, indicating that the instrumental variables were identifiable [60]. The weak identification tests (the Cragg–Donald Wald F statistic and Kleibergen–Paap rk Wald F statistic) for Model (2) were 141.16 and 30.088, respectively, both of which were larger than the Stock–Yogo weak ID test critical values at the 10% level of judgment (16.38). The weak identification tests (the Cragg–Donald Wald F statistic and Kleibergen–Paap rk Wald F statistic) for Model (4) were 72.765 and 17.389, respectively, both of which were larger than the Stock–Yogo weak ID test critical values at the 10% level of judgment (16.38), indicating that there was a strong correlation between the instrumental variables and the independent variable and that there was no weak instrumental variable problem [61].

Researchers have been working to determine the relationship between fulfilling environmental responsibilities and corporate performance; fulfilling environmental responsibilities contributes to their economic returns and encourages corporate managers to actively fulfill environmental responsibilities [62]. Researchers have also found that CER engagements help to maintain innovation performance and set goals to attract new global corporations to help businesses stay afloat [63]. In academia, how environmental investments affect various aspects of firm performance, such as environmental, operational, and economic performance, has been attracting increasing attention [64,65,66,67]. Few studies in the literature have directly studied the impact of corporate participation in environmental protection on sustainable development.

This study uses two indicators of environmental protection investment and environmental responsibility score to explain environmental protection behavior and the multi-faceted understanding of the impact of enterprises’ participation in environmental protection on sustainable development.

Previous studies on firms’ green behavioral decisions have mainly focused on the influence of government environmental regulations, market demand, public pressure, and firms’ profitability [68,69,70,71], but these studies have neglected the subjective initiative of firm management. This paper, however, from the perspective of managers’ cognition, discusses enterprises’ participation in environmental protection and further strengthens the research on the factors affecting the sustainable development of enterprises.

Previous studies have shown that government policies and regulations are the key to promoting corporate green investment [72,73] and improve corporate green technology innovation [74]. However, few studies start from investors themselves and explore how investors can promote corporate participation in environmental protection, thereby promoting corporate sustainable development. Therefore, this study provides a new idea for investors choosing what kind of enterprises to invest in.

## 5. Conclusions and Implications

### 5.1. Conclusions

Due to increasingly serious environmental issues, global attention has turned to enterprises, which are the main source of environmental pollution and also important participants in environmental protection. This study was based on 2011–2020 Chinese A-share listed companies. Using a fixed effects regression model, the relationship between enterprises participating in environmental protection and their sustainable development was examined. Environmental protection was analyzed from two aspects: enterprise performance responsibility and environmental protection investment. At the same time, green investors and green executive cognition were used as regulatory variables to evaluate their role in the relationship between environmental protection and the sustainable development of enterprises. The following conclusions were obtained from this study: First, environmental responsibility is affected by the sustainability of enterprises, and the performance of enterprises in carrying out environmental responsibilities can improve their sustainable development capacity. Second, environmental investment has a positive impact on enhancing enterprises’ capacity for continuous development. Enterprises’ environmental investments can effectively enhance their capacity for sustainable development. These two conclusions echo Moshirian et al.’s (2021) view [75], which suggests that environment protection is essential for the high-quality development of a country’s economy and has come to symbolize the capacity for a company’s sustainable development. Furthermore, previous studies have also shown enterprises increase investment in environmental protection, conscientiously fulfill their responsibilities for energy conservation, emission reduction, and environmental protection, and invest in environmental pollution control, which can send a positive message to society, help to speed up the establishment of a corporate image, and enhance consumers’ recognition of corporate products, which is similar to the basic conclusion.

Third, green executive cognition plays a positive role in environmental responsibility and environmental protection investment, thus affecting the sustainable development capacity of enterprises. There are few previous studies on executives’ green cognition, and they focus on researching executives’ professional background and green innovation [76]. Starting from the high-level ladder theory, this paper finds that executives’ green cognition and environmental protection careers can help increase corporate environmental protection behaviors, thereby promoting corporate sustainable development. Fourth, green investors play a positive role in environmental responsibility and protection, affecting the sustainable development capabilities of enterprises. Recent studies argue that investors systematically seek to include greener investments in their portfolios. From the perspective of institutional investors, this paper finds that green investors can promote enterprises to participate in environmental protection, increase the growth of corporate profits, and enrich the relevant theories of institutional investors.

### 5.2. Implications

First, enterprises should think deeply about their relationship with the natural environment and understand their responsibilities toward conserving resources and protecting the environment. Listed enterprises actively perform their environmental responsibilities. On the one hand, enterprises should actively bear environmental responsibility. In the process, enterprises can improve product quality, reduce agency costs between shareholders and managers, and improve their reputation. Policy support can help them obtain a strategic advantage through sustainable development and achieve a win–win situation of economic and environmental benefits. On the other hand, enterprises should attach importance to environmental protection input; proactively proceed from the source of production; increase investment and transformation in environmental protection technology, equipment, and other aspects; pursue clean production; reduce pollutant emissions; and improve their sustainable development capacity while protecting the environment. Finally, enterprises should strengthen their own perception of resource and environmental issues, enrich their knowledge of resources and the environment, improve their awareness of resource conservation and environmental protection responsibility, cultivate an awareness of green competition advantages among senior management, and establish a green enterprise image at the same time to attract green investors.

Second, executives play a decisive role in senior managers’ development strategies. Green cognition is the spiritual source of green development and a fundamental prerequisite for enterprises to implement green behavior. When executives have a higher level of understanding of the concept of green development, they will play a positive role in economic and environmental performance by participating in green governance to achieve a win–win situation. Therefore, high-level managers should improve their own environmental responsibility level and awareness of green competition advantages, create a green atmosphere in the enterprise, promote the participation of the enterprise in environmental protection, and enhance sustainable development.

Third, green investors should bear social and environmental responsibilities through targeted financial investments. Here, the investment goal is not only to realize economic benefits, but also to fulfill one’s social responsibility; investors can urge enterprises to improve their environmental performance and thus enhance corporate developmental capabilities. First, green investors should consider long-term investments. Research has found that the performance of earlier green funds in the market is better than that of those established later. Second is rational investment and the strengthening of the green investment concept. Third, risks should be rationally controlled. In addition, investors immensely benefit from green bonds, and a higher number of private equity bodies should be encouraged to enter the green market. This activity will also motivate industries and manufacturing units to use clean sources of energy in their production process. The expansion of the green bond market offers a viable option for enterprises and governments moving towards environmental protection [77].

Fourth, for the government, a combination of incentive tools could drive enterprises to participate in environmental protection. Due to the high cost and risk of green production, governance, and innovation, the government needs to further improve its various incentive policies while implementing relevant green laws and regulations. By providing environmental protection subsidies as well as tax reductions and exemptions to enterprises, the government can improve its enthusiasm for environmental protection investment and fulfill its environmental responsibility. However, the government should also vigorously advocate and publicize the concept of green development, encourage enterprises to pay attention to environmental protection, direct executives’ attention to the green expectations of stakeholders, attract green investors to the market, and realize the sustainable development of enterprises.

### 5.3. Limitations and Future Research

One limitation of this study is that it investigated only listed companies in Shanghai and Shenzhen, China, and the conclusions are not necessarily applicable to non-listed companies in China. Moreover, there was a lack of environmental investment data in corporate financial statements, so in future research, the lack of data should be addressed. Finally, this study mainly considered the impact of corporate environmental protection on the sustainable developmental capacity of enterprises, as well as green executive cognition and the regulation of green investors. In future research, it is necessary to conduct in-depth research on how the government can promote the sustainable development of enterprises. Government policies and regulations have played a positive role in promoting enterprises to participate in environmental protection. With the background of the “dual carbon” policy goal, enterprises should pay attention to switching to and upgrading their own low-carbon energy-saving measures, making good use of government environmental protection subsidies to realize clean and ecological production technology, and releasing positive energy to the outside world. The government should guide all sectors of society to attach importance to green governance and green investment, improve the environmental performance achieved by investment activities, improve the supervision mechanism for the use of environmental protection subsidies, implement earmarked funds, and organically combine government environmental protection subsidies and corporate environmental protection investment to jointly promote the green development of enterprises.

## Figures and Tables

**Figure 1 ijerph-20-04179-f001:**
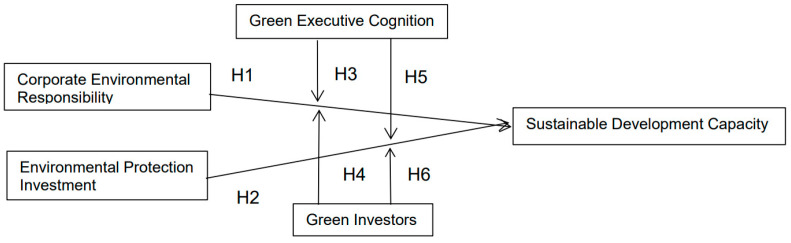
Research Model.

**Table 1 ijerph-20-04179-t001:** Definition and Measurement of the Variables.

Variable Type	Variable Name	Variable Symbol	Variable Measurement
Explained Variables	Sustainable Development Capacity	SGR	net sales interest rate × total asset turnover rate × income reservation rate × equity multiplication/(1 − net sales interest rate × total asset turnover rate × income reservation rate × equity multiplication)
Explanatory Variable	Enterprise Environmental Protection Investment	EPI	The natural logarithm of enterprise environmental protection investment
	Environmental Responsibility	CER	Hexun scoring
Moderating Variables	Green Executive Cognition	GEC	The logarithm of the frequency of the number of keywords appearing in the annual report of listed companies
	Green Investors	GI	Enterprises with green investors are scored 1, otherwise 0
Control Variables	Enterprise Size	SIZE	The natural logarithm of the total assets at the end of the year
	Asset-Liability Ratio	LEV	Total liabilities/total assets
	Cash Flow ratio	Cash Flow	Net cash flows from operating activities/total assets
	Corporate Growth	GROWTH	Operating income growth rate
	Total Asset Turnover Rate	ATO	Operating income/average total assets
	Firm Age	AGE	Ln (Present year−company founding year + 1)
	Year	YEAR	Annual dummy variables
	Industry	INDUSTRY	Industry dummy variable

**Table 2 ijerph-20-04179-t002:** Results of the descriptive statistics.

Variables	(1)	(2)	(3)	(4)	(5)	(6)	(7)	(8)
	N	Min	Max	Mean	Sd	P50	Skeness	Kurtosis
SGR	6456	−0.0167	0.266	0.0614	0.0518	0.0502	1.451	5.630
CER	6456	0	20	4.308	4.167	0	1.115	2.712
EPI	6456	0	21.65	15.70	3.356	14.71	−0.888	5.581
GI	6456	0	1	0.193	0.394	0	1.558	3.428
GEC	6456	0.693	6.098	3.326	0.993	3.332	0.097	3.397
Size	6456	19.52	26.40	22.45	1.286	22.29	0.635	3.166
Lev	6456	0.0310	0.906	0.432	0.195	0.429	0.075	2.193
ATO	6456	0.0563	2.902	0.906	0.407	0.996	0.230	4.325
Cashflow	6456	−0.200	0.257	0.0536	0.0629	0.0528	−0.092	4.224
FirmAge	6456	1.386	3.555	2.893	0.323	2.944	−0.863	4.238

**Table 3 ijerph-20-04179-t003:** Results of the correlation analysis.

	SGR	CER	EI	GI	EGP	Size	Lev	ATO	Cashflow	Firm Age
SGR	1									
CER	0.082 ***	1								
EPI	0.107 ***	0.038 ***	1							
GI	0.230 ***	−0.073 ***	0.135 ***	1						
EGP	0.067 ***	−0.070 ***	0.310 ***	0.154 ***	1					
Size	0.133 ***	0.118 ***	0.381 ***	0.221 ***	0.227 ***	1				
Lev	0.066 ***	0.085 ***	0.213 ***	0.033 ***	0.135 ***	0.561 ***	1			
ATO	0.220 ***	0.044 ***	−0.024 *	−0.024 *	−0.077 ***	−0.001	0.076 ***	1		
Cashflow	0.298 ***	−0.024 *	0.090 ***	0.077 ***	0.069 ***	0.052 ***	−0.159 ***	0.085 ***	1	
FirmAge	0.011	−0.121 ***	0.044 ***	0.045 ***	0.142 ***	0.182 ***	0.156 ***	0.001	0.082 ***	1

Notes: Robust *t*-statistics in parentheses. *** *p* < 0.01, * *p* < 0.1.

**Table 4 ijerph-20-04179-t004:** Results of the regression analysis.

Variables	Model 1	Model 2	Model 3	Model 4	Model 5	Model 6
	SGR	SGR	SGR	SGR	SGR	SGR
CER	0.0020 ***		0.0010 ***		0.0018 ***	
	(13.2696)		(3.5777)		(12.5658)	
EPI		0.0026 ***		0.0017 ***		0.0023 ***
		(14.2900)		(4.3820)		(11.8065)
EGP			0.0084 ***	0.0048 **		
			(6.5158)	(2.0590)		
GI					0.0246 ***	0.0109 **
					(14.5088)	(2.2744)
CER × EGP			0.0003 ***			
			(3.5835)			
EI × EGP				0.0002 **		
				(2.0692)		
CER × GI					0.0022 ***	
					(4.1630)	
EI × GI						0.0008 ***
						(2.8482)
Size	0.0110 ***	0.0109 ***	0.0101 ***	0.0101 ***	0.0070 **	0.0071 ***
	(3.7887)	(3.7992)	(3.5341)	(3.5585)	(2.5677)	(2.5972)
Lev	−0.0002	0.0006	0.0016	0.0026	0.0042	0.0048
	(−0.0181)	(0.0734)	(0.1931)	(0.3096)	(0.5184)	(0.5716)
ATO	0.0776 ***	0.0785 ***	0.0777 ***	0.0788 ***	0.0747 ***	0.0760 ***
	(12.9563)	(12.8897)	(13.0711)	(13.0308)	(13.2331)	(13.1793)
Cashflow	0.1425 ***	0.1436 ***	0.1415 ***	0.1423 ***	0.1381 ***	0.1388 ***
	(11.2729)	(11.2863)	(11.2911)	(11.2848)	(11.4640)	(11.3461)
FirmAge	−0.0176	−0.0118	−0.0217 *	−0.0151	−0.0226 *	−0.0156
	(−1.2990)	(−0.8588)	(−1.6509)	(−1.1195)	(−1.7633)	(−1.1770)
Constant	−0.2285 ***	−0.2838 ***	−0.2239 ***	−0.2693 ***	−0.1306 *	−0.1884 ***
	(−3.2083)	(−3.8947)	(−3.2129)	(−3.7355)	(−1.9473)	(−2.6825)
Hausman test *p*-value	0.0000	0.0000	0.0000	0.0000	0.0000	0.0000
Observations	6456	6456	6456	6456	6456	6456
R-squared	0.2328	0.2268	0.2425	0.2355	0.2791	0.2656
Industry FE	YES	YES	YES	YES	YES	YES
Year FE	YES	YES	YES	YES	YES	YES

Notes: Robust *t*-statistics in parentheses. *** *p* < 0.01, ** *p* < 0.05, * *p* < 0.1.

**Table 5 ijerph-20-04179-t005:** Robustness test.

Variables	Model 1	Model 2	Model 3	Model 4
	CER	SGR	EPI	SGR
	OLS first stage	OLS second stage	OLS first stage	OLS second stage
L.CER	0.2182 ***			
	(5.4875)			
CER		0.0042 ***		
		(5.2335)		
L.EPI			0.2093 ***	
			(4.1728)	
EPI				0.0055 ***
				(2.9912)
Size	0.6988 ***	0.0144 ***	0.4227	0.0155 ***
	(3.3030)	(3.8369)	(1.4064)	(4.1117)
Lev	0.4380	−0.0091	−0.5568	−0.0048
	(0.6117)	(−0.7736)	(−0.7320)	(−0.4040)
ATO	0.1711	0.0796 ***	−0.2533	0.0820 ***
	(0.4627)	(9.0191)	(−0.9425)	(9.2796)
Cashflow	0.5674	0.1454 ***	0.3237	0.1461 ***
	(0.5285)	(8.4716)	(0.3772)	(8.4736)
FirmAge	0.2983	−0.0288	−1.9074	−0.0140
	(0.1197)	(−1.3948)	(−1.3865)	(−0.6719)
Constant	−17.2438 **	−0.2562 **	10.6037	−0.4296 ***
	(−2.1990)	(−2.4642)	(1.3120)	(−4.1938)
Observations	3845	3845	3845	3845
R-squared	0.2009	0.2073	0.0459	0.2035
Industry FE	YES	YES	YES	YES
Year FE	YES	YES	YES	YES

Notes: Robust *t*-statistics in parentheses. *** *p* < 0.01, ** *p* < 0.05.

## Data Availability

The raw data supporting the conclusions of this article will be made available by the authors without undue reservation.

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
