# Peer review of "Corporate Environmental Protection Behavior and Sustainable Development: The Moderating Role of Green Investors and Green Executive Cognition"

_ijerph, 2023, doi:10.3390/ijerph20054179_

Round 1

Reviewer 1 Report

The authors of the paper "Corporate environmental protection behavior and sustainable development: The moderating role of green investors and green executive cognition" present a topic relevant to the context act, namely "if the environmental protection behavior of enterprises has a positive impact on their sustainable development...", which makes the work have a multiplying effect at the level of the business environment and from other states than the one analyzed by the authors of the work. Concepts, bibliographic sources, and citations are adequately mentioned within the paper, such as for example the works cited by the authors [18, 19, 20]"... among organizational policies are more inclined to implement ecological practices to achieve their sustainability objectives of the environment". Tables are properly referenced according to academic standards, but figures/graphs are missing from the paper, but this is a minor shortcoming.

The research methodology is adequate, respectively the 6 hypotheses are structured in accordance with the analyzed topic and with clarity on the defined subjects. For example, "Hypothesis 4: Green investors play a positive adjustment role in the impact of corporations, respectively in the evaluation of expenses for environmental protection according to the sustainable development capacity of enterprises" has structured in the formulation mode both the concept and the direct orientation towards the result . The data used are relevant and refer to Chinese companies listed with A shares from 2011 to 2020. The authors of the paper use the two models and verify the hypotheses of the regression model based on the fixed effects of CER and environmental investments on the sustainability of the enterprise.

The results of the paper are adequately presented from the point of view of the fact that this paper contributes to "the enrichment of the specialized literature on the environment, the mental protection behavior of enterprises and the sustainable development of enterprises and provides a theoretical basis for related research". However, we suggest the authors of the work to emphasize their personal scientific contributions, the results of the study, to the specialized scientific literature, as well as the elements of innovation that support the multiplying effect of scientific research.

The conclusions presented by the authors of the work capture the applicative elements of the work, namely "the listed companies actively fulfill their environmental responsibility". Moreover, the authors of the paper present the limitations of the study and the implications of the research team in the continuation of research in the specialized scientific field, but as we mentioned above, we suggest the authors of the paper to highlight their personal scientific and innovative contributions, for the specialized literature and not only from the point of view from an applicative point of view.

We congratulate the research team for the topic of the completed work, and after revision according to the aspects mentioned above, including the restructuring of the work, we propose the work to the editorial team for acceptance.

Author Response

  1. The authors of the paper "Corporate environmental protection behavior and sustainable development: The moderating role of green investors and green executive cognition" present a topic relevant to the context act, namely "if the environmental protection behavior of enterprises has a positive impact on their sustainable development...", which makes the work have a multiplying effect at the level of the business environment and from other states than the one analyzed by the authors of the work. Concepts, bibliographic sources, and citations are adequately mentioned within the paper, such as for example the works cited by the authors [18, 19, 20]"... among organizational policies are more inclined to implement ecological practices to achieve their sustainability objectives of the environment". Tables are properly referenced according to academic standards, but figures/graphs are missing from the paper, but this is a minor shortcoming.

Reply: According to the referee’s comments, we have added the research model (figure 1) and Skewness and kurtosis values (figure 2) in the study. (P6&P10)

Figure 1 is the research model.

Figure 1: Research Model

Figure 2: Skewness and kurtosis values

2.The research methodology is adequate, respectively the 6 hypotheses are structured in accordance with the analyzed topic and with clarity on the defined subjects. For example, "Hypothesis 4: Green investors play a positive adjustment role in the impact of corporations, respectively in the evaluation of expenses for environmental protection according to the sustainable development capacity of enterprises" has structured in the formulation mode both the concept and the direct orientation towards the result . The data used are relevant and refer to Chinese companies listed with A shares from 2011 to 2020. The authors of the paper use the two models and verify the hypotheses of the regression model based on the fixed effects of CER and environmental investments on the sustainability of the enterprise.

The results of the paper are adequately presented from the point of view of the fact that this paper contributes to "the enrichment of the specialized literature on the environment, the mental protection behavior of enterprises and the sustainable development of enterprises and provides a theoretical basis for related research". However, we suggest the authors of the work to emphasize their personal scientific contributions, the results of the study, to the specialized scientific literature, as well as the elements of innovation that support the multiplying effect of scientific research.

Reply: According to the referee’s comments, we have added the contents in the results of the paper as follows: (p13)

Researchers have been working to the relationship between fulfilling environmental responsibilities and corporate performance, fulfilling environmental responsibilities contributes to their economic returns, and encourages corporate managers to actively fulfill environmental responsibilities[63].The researchers also found that the CER engagements help to maintain innovation performance and set goals to attract new global corporations to help businesses stay afloat[64].In academia, how environmental investments affect various aspects of firm performance, such as environmental, operational, and economic performance, is attracting increasing attention[65,66,67,68].Few literatures directly study the impact of corporate participation in environmental protection on sustainable development.

This study uses two indicators of environmental protection investment and environmental responsibility score to explain environmental protection behavior, and the multi-faceted understanding of the impact of enterprises' participation in environmental protection on sustainable development.

Previous studies on firms’ green behavioral decisions have mainly focused on the influence of government environmental regulations, market demand, public pressure, and firms’ profitability [69,70,71,72], but these studies have neglected the subjective initiative of firm management. This paper, however, from the perspective of managers' cognition, this study discusses the enterprise's participation in environmental protection and further strengthens the research on the factors affecting the sustainable development of enterprises.

Previous studies have shown that government policies and regulations are the key to promoting corporate green investment[73,74], and improve corporate green technology innovation[75]. but few studies start from investors themselves and explore how investors can promote corporate participation in environmental protection, thereby promoting corporate sustainable development. Therefore, this study provides a new idea for investors to invest in what kind of enterprises to formulate.

3.The conclusions presented by the authors of the work capture the applicative elements of the work, namely "the listed companies actively fulfill their environmental responsibility". Moreover, the authors of the paper present the limitations of the study and the implications of the research team in the continuation of research in the specialized scientific field, but as we mentioned above, we suggest the authors of the paper to highlight their personal scientific and innovative contributions, for the specialized literature and not only from the point of view from an applicative point of view.

Reply: According to the referee’s comments, we have added the contents in the conclusions of the paper as follows: (p14)

These two conclusion echoes Moshirian et al.’s (2021) view [76], which suggests that environment protection are essential for the high-quality development of a country’s economy , and come to symbolize the capacity for a company’s sustainable development .Furthermore, previous study has also shown enterprises increase investment in environmental protection, conscientiously fulfill their responsibilities for energy conservation, emission reduction and environmental protection, and invest in environmental pollution control, which can send a positive message to the society, help to speed up the establishment of corporate image, and enhance consumers' recognition of corporate products, which is similar to the basic conclusion.

Third, Green executive cognition plays a positive role in environmental responsibility and environmental protection investment, thus affecting the sustainable development capacity of enterprises. There are few previous studies on executives' green cognition.and they focused on the research on executives' professional background and green innovation[77].Starting from the high-level ladder theory, this paper finds that executives' green cognition and environmental protection career can help increase corporate environmental protection behaviors, thereby promoting corporate sustainable development. Forth, Green investors play a positive role in environmental responsibility and protection, affecting the sustainable development capabilities of enterprises. Recent studies argue that investors systematically seek to include greener investments in their portfolios. From the perspective of institutional investors, this paper finds that green investors can promote enterprises to participate in environmental protection, increase the growth of corporate profits, and enrich the relevant theories of institutional investors.

4.We congratulate the research team for the topic of the completed work, and after revision according to the aspects mentioned above, including the restructuring of the work, we propose the work to the editorial team for acceptance.

Reply: Thank you for your comments and contributions.

Reviewer 2 Report

Title: Corporate environmental protection behavior and sustainable development: The moderating role of green investors and green executive cognition. 

I appreciate the chance to serve as a reviewer on this paper. The paper is well written and suits the scope of the issue. I urge the author(s) to consider the following comments and improve the paper accordingly.

11. At first, I think the authors should make clearer the economic meaning of their research starting with an economic motivation.

12. The authors should explain with accuracy (with a relative analysis) why their findings have implications for policy analysis (see for instance and make a comparative analysis the relative research by Tsagkanos et al. 2022).

23. It would also be useful for the audience and future researchers if a guide for the future research is provided: how this research could be used concretely to open new pathways? Is it possible to provide some examples and possible directions for future research?  

Literature

Tsagkanos A.,Sharma A., Ghosh B. (2022) “Green Bonds and Commodities: A new asymmetric sustainable relationship” Sustainability MDPI. 14, 6852.

Author Response

Title: Corporate environmental protection behavior and sustainable development: The moderating role of green investors and green executive cognition. 

I appreciate the chance to serve as a reviewer on this paper. The paper is well written and suits the scope of the issue. I urge the author(s) to consider the following comments and improve the paper accordingly.

  1. At first, I think the authors should make clearer the economic meaning of their research starting with an economic motivation.

Reply: According to the referee’s comments, we have added the contents as follows: (P1)

To resolve the contradiction between environmental protection and economic development, enterprises must focus on environmental protection. Corporate environmental behavior plays a vital role in achieving the dual goals of improving environmental quality and promoting sustainable economic development.

As environmental sustainability is becoming increasingly important for economic development, many Chinese companies have begun to implement environmental protection measures.

  1. The authors should explain with accuracy (with a relative analysis) why their findings have implications for policy analysis (see for instance and make a comparative analysis the relative research by Tsagkanos et al. 2022).

Reply: According to the referee’s comments, we have added the contents as follows: (P15)

In addition, investors are immensely beneficious of green bonds, a higher number of private equity bodies will be encouraged to enter the green market. This activity will also motivate industries and manufacturing units to use clean sources of energy in their production process. the expansion of the green bond market offers a viable perspective for enterprises and government towards environmental protection[78].

  1. It would also be useful for the audience and future researchers if a guide for the future research is provided: how this research could be used concretely to open new pathways? Is it possible to provide some examples and possible directions for future research?  

Literature

Tsagkanos A.,Sharma A., Ghosh B. (2022) “Green Bonds and Commodities: A new asymmetric sustainable relationship” Sustainability MDPI14, 6852.

Reply: According to the referee’s comments, we have added the contents as follows: (P16)

In future research, it is necessary to conduct in-depth research on how the government can promote the sustainable development of enterprises. Government policies and regulations have played a positive role in promoting enterprises to participate in environmental protection. Under the background of "dual carbon", enterprises should pay attention to their own low-carbon energy-saving transformation and upgrading, make good use of government environmental protection subsidies to realize clean and ecological production technology, and release positive energy to the outside world. The government guides all sectors of society to attach importance to green governance and green investment, improve the environmental performance achieved by investment activities, improve the supervision mechanism for the use of environmental protection subsidies, realize earmarked funds, and organically combine government environmental protection subsidies and corporate environmental protection investment to jointly promote the green development of enterprises.

Reviewer 3 Report

The paper is well written, and the topic is interesting, so before acceptance of this paper, I have some major suggestions:

 1. The introduction must be added one paragraph about this study's main motivation and significance.

2. On Page# 8,  line #314 (3.3 Model design), it is suggested to combine all equations in one table or mention them together and then explain all models at once because it does not look professionally attractive.

3. In Table#2, line# 359, I have some concerns about descriptive statistics, especially "mean and SD values," i.e., GEC, etc., so it is suggested to revisit them and include skewness and kurtosis values as a clear picture of the sample distribution.

4. Further in the robustness section, Table 5 also suggested incorporating the results with three-stage least squares (3SLS) as it is now widely used in such studies.

5. Finally, I found no article cited from 2023. The authors can search by themselves or may take help from these relevant studies**

 **a). Li, Y., Hu, S., Zhang, S., & Xue, R. (2023). The Power of the Imperial Envoy: The Impact of Central Government Onsite Environmental Supervision Policy on Corporate Green Innovation. Finance Research Letters, 52, 103580. https://doi.org/10.1016/j.frl.2022.103580

b). Cheng, Z., & Yu, X. (2023). Can central environmental protection inspection induce corporate green technology innovation?. Journal of Cleaner Production, 135902. https://doi.org/10.1016/j.jclepro.2023.135902

c). Sabbir, M. M., & Taufique, K. M. R. (2023). Sustainable employee green behavior in the workplace: Integrating cognitive and noncognitive factors in corporate environmental policy. Business Strategy and the Environment, 31(1), 110-128. https://doi.org/10.1002/bse.2877

d). Hu, Y., Bai, W., Farrukh, M., & Koo, C. K. (2023). How does environmental policy uncertainty influence corporate green investments?. Technological Forecasting and Social Change, 189, 122330. https://doi.org/10.1016/j.techfore.2023.122330

e) Liu, Y., Liu, J., & Liu, S. (2023). Executive Team Functional Background and Enterprise Green Technology Innovation. books.google.com

Author Response

The paper is well written, and the topic is interesting, so before acceptance of this paper, I have some major suggestions:

  1. The introduction must be added one paragraph about this study's main motivation and significance.

 Reply: According to the referee’s comments, we have added the contents as follows: (P3)

The study aims to analyze the behavior of enterprises participating in environmental protection from two aspects: CER and environmental protection investment. This study explores the role of corporate environmental protection behavior in achieving the dual goals of improving environmental quality and promoting sustainable economic development.

The contribution point of this study is that:First, this study This study divides enterprise environmental protection behavior into two parts: environmental investment and environmental responsibility, enriches the literature on the environmental protection behavior of enterprises and the sustainable development of enterprises and provides a theoretical foundation for research in related fields. Second, this study explores the role of green executives' cognition in promoting the relationship between environmental protection and corporate sustainable development from within the enterprise, and provides insights into how top managers within the enterprise can promote the green behavior of employees and embed green culture in the entire organization; from the outside of the enterprise, the entry of green investors has increased. The behavior of enterprises to participate in environmental protection. Government policies and regulations are key to driving corporate green investment, so this study provides new insights into the formulation of environmental policies. Third, the research results of this study provide important practical implications for companies to design sustainable development strategies. In the long run, the environmental protection behavior of enterprises can improve environmental quality, promote economic development, enhance enterprise value, and help enterprises establish long-term competitiveness.

  1. On Page# 8, line #314 (3.3 Model design), it is suggested to combine all equations in one table or mention them together and then explain all models at once because it does not look professionally attractive.

 Reply: According to the referee’s comments, we have combine all equations in on table: (P9)

Table 2. Models

                                                                                           (1)

                                                                                          (2)

                                                                                 (3)

                                                                                 (4)

                                                                                 (5)

                                                                                 (6)

  1. In Table#2, line# 359, I have some concerns about descriptive statistics, especially "mean and SD values," i.e., GEC, etc., so it is suggested to revisit them and include skewness and kurtosis values as a clear picture of the sample distribution.

Reply: We found that some descriptive statistics are typed incorrect in Table#2 of first submitted paper. For example, the values of SD and P50 are the same in the first version. So, we have revised and include skewness and kurtosis values as a clear picture of the sample distribution. (P9&P10)

Table 2. Results of the descriptive statistics

(1)

(2)

(3)

(4)

(5)

(6)

VARIABLES

N

min

max

mean

sd

P50

SGR

6,456

-0.0167

0.266

0.0614

0.0518

0.0502

CER

6,456

0

20

4.308

4.167

0

EPI

6,456

0

21.65

15.70

3.356

14.71

GI

6,456

0

1

0.193

0.394

0

GEC

6,456

0.693

6.098

3.326

0.993

3.332

Size

6,456

19.52

26.40

22.45

1.286

22.29

Lev

6,456

0.0310

0.906

0.432

0.195

0.429

ATO

6,456

0.0563

2.902

0.906

0.407

0.996

Cashflow

6,456

-0.200

0.257

0.0536

0.0629

0.0528

FirmAge

6,456

1.386

3.555

2.893

0.323

2.944

Figure 2: Skewness and kurtosis values

  1. Further in the robustness section, Table 5 also suggested incorporating the results with three-stage least squares (3SLS) as it is now widely used in such studies.

Reply: As far as we know, three-stage least squares (3SLS) is that two-stage least squares (2SLS) adds seemingly unrelated regression(SUR). The process of estimating the correlation between different equations is the same as SUR, Using 2SLS to solve the problem of coalescence.

SUR is a name that appears to be unrelated but is actually related. The SUR model is used when there are many dependent variables and there is a correlation between dependent variables.

However, this study conducted a fixed regression analysis using one dependent variable called SGR to verify each hypothesis. In addition, in order to control endogenous problems, a robust test was additionally conducted using the 2sls model. Therefore, it is judged that 3sls is not a suitable model for this study, and 2sls is the most suitable model.

  1. Finally, I found no article cited from 2023. The authors can search by themselves or may take help from these relevant studies**

 **a). Li, Y., Hu, S., Zhang, S., & Xue, R. (2023). The Power of the Imperial Envoy: The Impact of Central Government Onsite Environmental Supervision Policy on Corporate Green Innovation. Finance Research Letters, 52, 103580. https://doi.org/10.1016/j.frl.2022.103580

b). Cheng, Z., & Yu, X. (2023). Can central environmental protection inspection induce corporate green technology innovation?. Journal of Cleaner Production, 135902. https://doi.org/10.1016/j.jclepro.2023.135902

c). Sabbir, M. M., & Taufique, K. M. R. (2023). Sustainable employee green behavior in the workplace: Integrating cognitive and non‐cognitive factors in corporate environmental policy. Business Strategy and the Environment, 31(1), 110-128. https://doi.org/10.1002/bse.2877

d). Hu, Y., Bai, W., Farrukh, M., & Koo, C. K. (2023). How does environmental policy uncertainty influence corporate green investments?. Technological Forecasting and Social Change, 189, 122330. https://doi.org/10.1016/j.techfore.2023.122330

  1. e) Liu, Y., Liu, J., & Liu, S. (2023). Executive Team Functional Background and Enterprise Green Technology Innovation. books.google.com

Reply: According to the referee’s comments, we have added all the references as follows:

[46] Sabbir, M. M., & Taufique, K. M. R. (2023). Sustainable employee green behavior in the workplace: Integrating cognitive and non‐cognitive factors in corporate environmental policy. Business Strategy and the Environment, 31(1), 110-128. https://doi.org/10.1002/bse.2877

[73] Li, Y., Hu, S., Zhang, S., & Xue, R. (2023). The Power of the Imperial Envoy: The Impact of Central Government Onsite Environmental Supervision Policy on Corporate Green Innovation. Finance Research Letters, 52, 103580. https://doi.org/10.1016/j.frl.2022.103580

[74]Hu, Y., Bai, W., Farrukh, M., & Koo, C. K. (2023). How does environmental policy uncertainty influence corporate green investments?.Technological Forecasting and Social Change, 189, 122330. https://doi.org/10.1016/j.techfore.2023.122330

[75]Cheng, Z., & Yu, X. (2023). Can central environmental protection inspection induce corporate green technology innovation?. Journal of Cleaner Production, 135902. https://doi.org/10.1016/j.jclepro.2023.135902

[77]Liu, Y., Liu, J., & Liu, S. (2023). Executive Team Functional Background and Enterprise Green Technology Innovation. books.google.com

Round 2

Reviewer 3 Report

Well, I completed satisfied with the changes done by authors but just one minor suggestion.

1. The Skewness and kurtosis values should be shown in descriptive stats Table#2, no need to shown in graphical form.

 Congradulates authors.

Author Response

Well, I completed satisfied with the changes done by authors but just one minor suggestion.

  1. The Skewness and kurtosis values should be shown in descriptive stats Table#2, no need to shown in graphical form.

 Congradulates authors.

Reply: Thank you for reviewer’s comments. We have added the skewness and kurtosis values in Table#2. We have used the STATA Statistics Program. (P9-P10)

Table 2. Results of the descriptive statistics

(1)

(2)

(3)

(4)

(5)

(6)

(7)

(8)

VARIABLES

N

min

max

mean

sd

P50

skeness

kurtosis

SGR

6,456

-0.0167

0.266

0.0614

0.0518

0.0502

1.451

5.630

CER

6,456

0

20

4.308

4.167

0

1.115

2.712

EPI

6,456

0

21.65

15.70

3.356

14.71

-0.888

5.581

GI

6,456

0

1

0.193

0.394

0

1.558

3.428

GEC

6,456

0.693

6.098

3.326

0.993

3.332

0.097

3.397

Size

6,456

19.52

26.40

22.45

1.286

22.29

0.635

3.166

Lev

6,456

0.0310

0.906

0.432

0.195

0.429

0.075

2.193

ATO

6,456

0.0563

2.902

0.906

0.407

0.996

0.230

4.325

Cashflow

6,456

-0.200

0.257

0.0536

0.0629

0.0528

-0.092

4.224

FirmAge

6,456

1.386

3.555

2.893

0.323

2.944

-0.863

4.238
